# Detection of Chymotrypsin by Optical and Acoustic Methods

**DOI:** 10.3390/bios11030063

**Published:** 2021-02-26

**Authors:** Ivan Piovarci, Tibor Hianik, Ilia N. Ivanov

**Affiliations:** 1Department of Nuclear Physics and Biophysics, Faculty of Mathematics, Physics and Informatics, Comenius University, Mlynska Dolina F1, 842 48 Bratislava, Slovakia; piovarci6@uniba.sk; 2Center for Nanophase Materials Sciences, Oak Ridge National Laboratory, Oak Ridge, TN 37831, USA; ivanovin@ornl.gov

**Keywords:** chymotrypsin, β-casein, nanoparticles, UV-vis spectroscopy, dynamic light scattering, quartz crystal microbalance

## Abstract

Chymotrypsin is an important proteolytic enzyme in the human digestive system that cleaves milk proteins through the hydrolysis reaction, making it an interesting subject to study the activity of milk proteases. In this work, we compared detection of chymotrypsin by spectrophotometric dynamic light scattering (DLS) and quartz crystal microbalance (QCM) methods and determined the limit of chymotrypsin detection (LOD), 0.15 ± 0.01 nM for spectrophotometric, 0.67 ± 0.05 nM for DLS and 1.40 ± 0.30 nM for QCM methods, respectively. The sensors are relatively cheap and are able to detect chymotrypsin in 3035 min. While the optical detection methods are simple to implement, the QCM method is more robust for sample preparation, and allows detection of chymotrypsin in non-transparent samples. We give an overview on methods and instruments for detection of chymotrypsin and other milk proteases.

## 1. Introduction

Proteases represent a very wide and important group of enzymes found in a broad range of biological systems [1]. Proteases play an important role in the digestion process and participate in various pathological processes [2,3]. Chymotrypsin is a serine protease present in the human digestive system that participates in protein cleavage in the intestines [4]. Together with trypsin, chymotrypsinogen is ejected into the duodenum, where trypsin cleaves it into the active form [5]. Chymotrypsin activity is closely related to the activity of trypsin, which, along with plasmin, is an important enzyme in milk. Activity of plasmin is correlated to the quality of milk where the protease cleaves the proteins, mainly casein micelles affecting the milk flavor, shelf-life or cheese yield [6]. In pathology and medicine, chymotrypsin also has anti-inflammatory effects and has been successfully used to reduce post-operation complications after cataract surgery [7]. Measuring chymotrypsin activity can also be used for differential diagnosis [8].

Thus, development of sensitive, inexpensive, fast, and easy to use methods for detection of chymotrypsin or other milk proteases would be beneficial to disease diagnostics and control of dairy quality. However, there are no simple and effective assays that can be used for these purposes yet available. Protease detection is currently based on the detection of α-amino groups cleaved from the protein substrate using optical or high-performance liquid chromatography (HPLC) methods. The method that can be used for fast analysis of the protease concentration is based on enzyme-linked immunosorbent assay (ELISA) with a limit of detection (LOD) of about 0.5 nM for chymotrypsin [9,10]. However, the above-mentioned methods do not allow study of the kinetics of substrate digestion.

In this paper we test three methods for chymotrypsin detection: QCM, spectrophotometric, and DLS. 

The QCM method is based on measurement of the resonant frequency, f, of shearing oscillations of AT-cut quartz crystal, as well as motional resistance, R_m_, and is also known as thickness shear mode method (TSM). The protease substrates, such as β-casein or short specific peptides, are immobilized on thin gold layers sputtered at a QCM transducer. High frequency voltage, typically in the range of 5–20 MHz, induces shearing oscillations of the crystal. The fundamental resonance frequency of the crystal, f_0_, depends on the physical properties of the quartz viscosity of the medium to which the crystal surface is exposed, as well as on the molecular interactions at the surface. The R_m_ value is sensitive to shearing viscosity, which is due to the molecular slip between the protein layer and surrounding water environment. Using Sauerbrey Equation (1) [11], one can link the change in resonant frequency to the mass bound to the surface of the electrode.
Δf = −2f_o_^2^Δm/A(μ_q_ρ_q_)^1/2^,(1)
where f_o_ is the fundamental resonant frequency (Hz), A is the active crystal area (in our case: 0.2 cm^2^), ρ_q_ is quartz density (2.648 g cm^−3^), Δm is the mass change (g), ρ_q_ is the shear modulus of the crystal (2.947 × 10^11^ g cm^−1^ s^−2^). This Equation is valid only for a rigid layer in vacuum. In a liquid environment and for relatively soft layers, the viscosity contribution can be estimated by measurements of R_m_.

We modified the surface of the QCM crystal with a layer of β-casein. The resulting mass added to the sensor leads to the decrease of the resonant frequency, f, and increase of motional resistance, R_m_. Chymotrypsin will cleave β-casein, which results in an increase in f and decrease in R_m_ values. The mass sensitive QCM method was used for the detection of trypsin activity using synthesized peptide chains [12]. Poturnayova et al. used β-casein layers to detect activity of plasmin and trypsin with LOD around 0.65 nM [13]. Incorporation of machine learning algorithm for analysis of multiharmonic QCM response allowed detection of trypsin and plasmin with LOD of 0.2 nM and 0.5 nM, respectively. The applied algorithm in the work of Tatarko et al. allowed us to distinguish these two proteases within 2 min [14].

We also used the spectrophotometric method based on measurement of absorbance of the dispersion of gold nanoparticles (AuNPs) coated by 6-mercapto-1-hexanol (MCH) and β-casein. AuNPs demonstrate a surface plasmon resonance (SPR) effect, which arises from the oscillating electromagnetic field of light rays getting into contact with the free electrons in metallic nanoparticles and induces their coherent oscillation, which have strong optical absorption in the UV-vis part of the spectrum. The SPR absorbance of AuNPs depends on the surrounding medium and on the distance between nanoparticles [15]. In the work by Diouani, AuNPs modified with casein were used to detect Leishmania infantum using amperometric methods [16]. Chen et al. modified AuNPs with a trypsin-specific peptide sequence [17]. After the trypsin cleavage, the gold nanoparticles aggregated, which was detected by monitoring changes in the UV-vis spectrum. The detection limit of this method was estimated to be around 5 nM. Svard et al. modified gold nanoparticles with casein or IgG antibodies for detection trypsin or gingipain activity, by measuring SPR peak shift (blue shift for trypsin and red shift for gingipain) and reporting LOD of less than 4.3 nM for trypsin and gingipain [18]. Goyal et al. developed method of immobilization of gold nanoparticles on a paper membrane [19]. The protease activity then led to aggregation of the gold nanoparticles on the membrane and resulted in a colorimetric response in a visible part of the spectrum detectable by the naked eye. AuNPs modified by gelatin that served as a substrate for proteinase digestion have also been used for detection of other proteases such as trypsin and matrix metalloproteinase-2 [20]. In our work, we modified the gold nanoparticles with β-casein and MCH using protocol from Ref. [20]. The β-casein protects the AuNPs from aggregation. Addition of the chymotrypsin and subsequent cleavage of the β-casein caused nanoparticles aggregation due to loss of the protective shell. This effect was observed by measuring UV-vis spectra of nanoparticle dispersion.

We also used dynamic light scattering (DLS) method which uses Brownian motion and the Rayleigh scattering of the light from particles to assess their size [21]. The intensity of the scattered light (which depends on particle concentration) changes over time because of particle aggregation. The auto-correlation function that correlates the intensity of scattered light with its intensity after an arbitrary time is used to discern the size of the particles. The auto-correlation function also depends on diffusion coefficient of the nanoparticles [22]. In DLS experiments we used AuNPs modified with β-casein. After addition of the chymotrypsin, we were able to observe the cleavage of the casein layer without AuNPs aggregation that resulted in a decrease of the size of nanoparticles.

This report is an extension of a manuscript published in proceeding of the 1st International Electronic Conference on Biosensors [23].

## 2. Materials and Methods

### 2.1. Chemicals

Auric acid (HAuCl_4_), sodium citrate, β-casein (Cat. No. C6905), 6-mercapto-1-hexanol (MCH, Cat. No. 725226), phosphate buffered saline (PBS) tablets (Cat. No. P4417), 11-mercaptoundecanoic acid (MUA, Cat. No. 450561), *N*-(3-Dimetylaminopropyl)-*N*′-etylcarbodiimid (EDC, Cat. No. E6383), *N*-Hydroxisuccinimid (NHS, Cat. No. 130672) and α-chymotrypsin (Cat. No. C3142) were of highest purity and purchased from Sigma-Aldrich (Darmstadt, Germany). Standard chemicals (p.a. grade), NaOH, HCl NaOH, NH_3_, and H_2_O_2_ were from Slavus (Bratislava, Slovakia). Deionized water was prepared by Purelab Classic UV (Elga, High Wycombe, UK).

### 2.2. Spectrophotometric UV-vis Method

Gold nanoparticles (AuNPs) were prepared by modified citrate method [24]. In short, 100 mL of 0.01% chloroauric acid (HAuCl_4_) was heated at around 98 °C and then 5 mL of 1% sodium tris-citrate was added. This solution was maintained at the temperature 98 °C and stirred by magnetic stirrer until it turned deep red (for about 15 min). Then the solution of AuNPs was cooled down and stored in the dark. To modify the gold nanoparticles with β-casein, we added 2 mL of 0.1 mg/mL aqueous β-casein into 18 mL of the AuNPs solution. After 2 h of incubation at room temperature without stirring, the gold nanoparticles were further incubated with 200 µL of 1 mM MCH overnight for approximately 18 h. MCH removes the surface charge of nanoparticles and thus facilitates their aggregation [20]. This is reflected by a color change to violet. However, nanoparticles (NPs) are protected from full aggregation due to the presence of a β-casein layer. Addition of chymotrypsin caused cleavage of β-casein, and as a result, the NPs aggregate. This was reflected by changes of the color of the solution to blue and then it became colorless. For the experiments, we prepared 0.95 mL of NPs. Chymotrypsin was dissolved in deionized water and 0.05 mL of chymotrypsin from the stock solution (concentration 100 nM) was added to each cuvette (1 mL standard cuvette, type UV transparent, Sarstedt, Nümbrecht, Germany). The concentration of chymotrypsin in cuvettes was 0.1, 0.3, 0.5, 0.7, 1; 5, and 10 nM at 1 mL of the total volume of solution. We also used a reference cuvette where only 0.05 mL of protease-free water was added to the AuNPs solution (total volume 1 mL). The spectra of the AuNPs were measured before protease addition (*t* = 0 min), just after protease addition (approximately 30 s) and then every 15 min up to 60 min. The measurements were repeated 3 times. The value of absorbance at time *t* = 0 has been multiplied by the dilution factor to correct the changes in absorbance intensity caused by the initial protease addition. Absorbance was measured by UV-1700 spectrophotometer (Shimadzu, Kyoto, Japan). The scheme of AuNPs modification and chymotrypsin cleavage is presented in Figure 1.

### 2.3. DLS Method

The AuNPs prepared as described in Section 2.2. were incubated with 0.1 mg/mL of aqueous β-casein solution overnight in a volume ratio of 1:9 of β-casein to AuNPs (MCH was not used in this case). Addition of the chymotrypsin to the solution of AuNPs modified by β-casein did not lead to any discernible color change; however, it was possible to detect the decreased size of the AuNPs using ZetaSizer Nano ZS (Malvern Instruments, Malvern, UK). First, the size of AuNPs was measured in 1 mL standard cuvette (1 mL standard cuvette, type UV transparent, Sarstedt, Nümbrecht, Germany). Then the 0.1 mL of water solution containing various chymotrypsin concentrations was added to each cuvette. The final concentration of chymotrypsin in cuvettes was 0.1, 0.3, 0.5, 0.7, 1; 5 and 10 nM at 1.1 mL total volume of solution. The size of nanoparticles was measured before addition of chymotrypsin (*t* = 0) right after the addition (approximately 1 min) and after 30 min.

### 2.4. Quartz Crystal Microbalance (QCM) Method

The acoustic QCM sensor was prepared using an AT-cut quartz piezocrystals (f_o_ = 8 MHz, ICM, Oklahoma, OK, USA) with sputtered thin gold layers of an area A = 0.2 cm^2^, that served as electrodes. First, the crystal was carefully cleaned as follows. It was exposed to a basic Piranha solution (H_2_O_2_:NH_3_:H_2_O = 1:1:5 mL). The crystals were immersed for 25 min in this solution, in beakers in a water bath (temperature was approximately 75 °C). Subsequently, the crystals were withdrawn, rinsed with distilled water, and returned to the beaker with a new dose of Piranha solution on the reverse side of the crystal. This was repeated three times. On the last extraction, the crystals were washed three times with distilled water and then with ethanol and placed in a bottle containing ethanol for storage at room temperature. The clean crystal was incubated overnight for 16–18 h at room temperature with 2 mM MUA dissolved in ethanol. MUA is a carboxylic acid with a sulfide group (SH). The sulfide moiety interacts with the gold on the crystal to form a self-assembled layer. After incubation, the crystal was washed with ethanol, distilled water, and 20 mM EDC and 50 mM NHS were applied for 25 min. These substances react with the carboxyl moiety of MUA and activate them to form a covalent bond with amino acids. Subsequently, the crystal was washed by distilled water, dried with nitrogen, and placed in an acrylic flow cell (JKU Linz, Austria). The cell was filled with PBS buffer using a Genie plus 2011 step pump (Kent Scientific, Torrington, CT, USA) at a flow rate of 200 μL/min. After filling the cell, we switched the flow to the rate of 50 μL/min. Then, 1 mg/mL of β-casein dissolved in PBS was allowed to flow under the crystal modified by MUA layer. After 35 min, only pure PBS was flowed in order to remove the unbound β-casein. All steps of the preparation of β-casein layer were recorded using a research quartz crystal microbalance (RQCM) instrument (Maxtek, East Syracuse, NY, USA).

After binding of β-casein to the electrode surface and stabilizing the resonant frequency (washing out all unbound residues), we applied chymotrypsin to the crystal at concentrations of 1 pM, 10 pM, 100 pM, 1 nM, 10 nM and 20 nM. After 35 min of chymotrypsin application, the PBS was let to flow into the cell until the resonant frequency stabilized. The change in casein coated QCM resonant frequency from application of chymotrypsin to stabilization in PBS corresponds to the amount of casein cleaved from the layer. After lower concentrations (1 pM, 10 pM, 100 pM), we applied a higher concentration of chymotrypsin (at least at a concentration 2 orders of magnitude higher). In such measurements, we analyzed the degree of cleavage as the change in frequency from the initial state to a steady-state value. All measurements were performed at PBS, pH 7.4. 

For optical and gravimetric methods, the limit of detection (LOD) was determined using following Equation:LOD = 3.3 × (SD)/S,(2)
where SD is standard deviation of the sample with lowest concentration and S is slope determined from fit of linear part of the calibration curve. The sequence of QCM operation including surface modification and sensing cleavage of β-casein by chymotrypsin using QCM piezocrystal and is presented in Figure 2.

## 3. Results and Discussion

### 3.1. Detection of Chymotrypsin by Optical Method

In the first series of experiments, we studied the cleavage of the β-casein at the surface of the AuNPs by UV-vis and DLS methods. AuNPs modified by β-casein and MCH were used in optical detection method. Figure 3 shows the change in the absorption spectra after each step of AuNPs modification. The modification of AuNPs with β-casein resulted in a shift of the maximum of absorbance by around 5 nm toward higher wavelengths and in a slight increase in absorbance. After addition of MCH which replaces the β-casein protective layer leads to broadening of absorption peak, and shifts by 60 nm toward higher wavelengths, indicating increase in size due to aggregation of AuNP. The results agree well with Ref. [20] for AuNPs modified by gelatin and MCH.

The changes of absorbance spectra of AuNPs suspension have been measured during the chymotrypsin cleavage at 0 min, 0.5 min,15 min, 30 min and 60 min. Changes in spectra over time for two different concentration of chymotrypsin are presented in Figure 4. At a relatively low concentration of chymotrypsin (0.1 nM), we did not observe significant changes of the absorbance (Figure 4a). However, at higher chymotrypsin concentration, around 10 nM, a substantial red shift of the spectra was observed (up to 615 nm). It can be also seen that after maximum shifting substantial decrease of the absorbance with time occurred.

Figure 5 shows the absorbance and change of maximum position of absorbance peak in time for all concentrations of chymotrypsin studied.

The rate of decrease in AuNP absorbance is higher for concentration of chymotrypsin 5 and 10 nM, and at lower concentration of chymotrypsin the rate of change is much slower (Figure 5a). The maximum position of absorbance peak shifted with time substantially at higher chymotrypsin concentrations (5 and 10 nM). For 5 and 10 nM chymotrypsin the maximum position of absorption peak was stabilized at around 615 nm, while for lower concentrations it increased with time almost linearly (Figure 5b). In order to prepare the calibration curve, we fit the change of absorbance vs. time with linear curve and then differentiated this model numerically to obtain the values of dA/dt. The calibration curve is presented on Figure 6a.

We were able to use reverse Michaelis—Menten model to analyze the obtained data. However, instead of substrate concentration, the concentration of chymotrypsin, c, has been used in this model: v = v_max_ [c/(K_M_ + c)], where v and v_max_ are the rate and maximum rate of enzyme reaction, respectively, and K_M_ = 3.89 ± 1.24 nM is reverse Michaelis—Menten constant obtained from the fit using the Michaelis—Menten model (R^2^ = 0.96) (Figure 6b). In our case, v = dA/dt and v_max_ = (5.3 ± 0.9) × 10^−3^ min^−1^. However, this model was used only formally because of different restraints. The main assumptions of the excess enzyme and limited substrate concentration was reversed in this case and instead, the concentration of the enzyme changed while the substrate was presented in excess. This implies different meaning of K_M_ (compared to the Michaelis—Menten model) which now represents the concentration of enzyme at which the rate of reaction is half of the maximum instead of concentration of substrate. A limitation of this approach is the assumption of substrate excess; nevertheless, it can be used for good approximation [25]. To calculate LOD, we used only part of the calibration curve from 0 to 5 nM, where the plot of dA/dt vs. c was almost linear. The results are shown in Figure 6b.

The LOD of the optical method of chymotrypsin detection, 0.15 ± 0.01 nM, was calculated from the Equation (2) using SD = 0.29 min^−1^ and S = 6.3 min^−1^ nM^−1^. This value is 3.3 times lower than that of the ELISA method reported in the literature, around 0.5 nM [10]. However, in contrast with ELISA which requires specific antibodies, the method based on AuNPs is much easier and faster. Detection time of chymotrypsin using the optical method is about 30 min. The detection of the chymotrypsin with β-casein and MCH modified AuNPs can be done in one step. The disadvantage of this method is that only transparent samples can be used for detection. This restriction can be lifted using the surface sensitive gravimetric method (see Section 3.3).

### 3.2. Detection of Chymotrypsin by DLS Method

We measured the Z-average size of the non-modified AuNPs, which was found to be around 20 nm. The size is bigger than the assumed size of the prepared AuNPs of around 15 nm [22]. This is explained by the fact that DLS technique tends to overestimate the size of the gold nanoparticles due to the hydration sphere around the AuNPs. The Z-average size of the AuNPs modified with β-casein was around 35 nm. Figure 7. shows the plot of the Z-average size of AuNPs modified by β-casein at time 0 and 30 min at presence of various concentrations of chymotrypsin. Incubation of AuNPs with chymotrypsin resulted in decrease of Z-average size, which is more remarkable at presence of 5 nM and 10 nM protease concentrations. The variation in AuNP size at time 0 is related to the original size of nanoparticles, as well as rather fast cleavage of casein by protease, especially at its higher concentrations. However, even at relatively high chymotrypsin concentrations (10 nM) the average size did not reach those of naked AuNPs. This is evidence that cleavage was not complete and there is still a residual β-casein layer around AuNPs. This also explains why incomplete aggregation was observed in UV-vis experiments.

We also constructed a calibration curve based on the percentual change of Z-average size in 30 min (Figure 8a) and fitted this by reverse Michaelis—Menten model. About 25% of the Z-average size was observed in 30 min. However, we can also see that the data has quite large (5 nm) standard deviation (obtained from 3 independent experiments at each concentration of chymotrypsin) affecting accuracy of concentration measurements, and the LOD of the sensor. Nevertheless, it is still a useful method to detect presence of chymotrypsin in the sample. The standard deviation could be improved by increasing number of measurements of the sample. It is important to note that the enzyme reaction was used without buffering the solution, which could also lead to a large value of standard deviation. The recommended buffer for chymotrypsin is 100 mM Tris-HCl at pH 7.8 (optimum pH) containing 10 mM CaCl_2_ for stability. However, since we observed AuNPs aggregation in buffer, the experiments were carried in un-buffered solution. The calibration curve seems to be saturated near the 10 nM chymotrypsin.

Using a Michaelis—Menten reverse model from the fit of the results presented on Figure 8a we obtained for K_M_ = 1.03 ± 0.26 nM (R^2^ = 0.998). This value is almost four times less than that obtained via spectrophotometric methods. This can be explained by addition of MCH in the spectrophotometric method, which can interfere with the cleavage of β-casein and decelerate the reaction. Both optical methods, however, should be able to detect the protease activity with similar precision. We took the linear part of the calibration curve (Figure 8b) and calculated LOD = 0.67 ± 0.05 nM. LOD value was calculated from the Equation (2) using SD = 2.54 (%) and S = 12.47 (%) nM^−1^. This value is 4.5 times higher in comparison with those obtained by spectrophotometric method. The possible reason is less reproducible data in the case of Z_average_ measurement in comparison with the absorbance method. The time of measurement is practically the same for both optical methods. In the case of the DLS method, the preparation of the AuNPs is by one step easier and since there is no MCH in the sample the AuNPs are more stable than those used in the spectrophotometric method.

### 3.3. Detection of Chymotrypsin by Gravimetric Method

For the gravimetric method, we first modified the surface of the QCM piezocrystal with MUA and then by β-casein. By monitoring resonant frequency, f, and motional resistance, R_m_, it was possible to study all steps of preparation of the sensing surface. The value of motional resistance reflects the viscosity contribution caused by non-ideal slip between β-casein-layer and the surrounding water environment [26]. This is presented in Figure 9. The activation of carboxylic groups of MUA with EDC/NHS lead to only a small shift in resonance frequency. The addition of β-casein resulted in a fast drop of resonant frequency of about 170 Hz. After changing the flow with buffer, the frequency increased due to the removal of nonspecifically bound β-casein. The resulting frequency shift after washing of the surface corresponded to 120 Hz. From the Sauerbrey equation, we can calculate the change in mass on QCM biosensor, which corresponded to about 165 ng of mass added. With the knowledge of the molecular weight of β-casein M_w_ = 24 kDa, we could calculate the surface density of β-casein: Г = 34.5 pM/cm^2^. From the changes in motional resistance R_m_, we could estimate the contribution of surface viscosity into resonant frequency. Since the R_m_ value decreases and increases proportionally to the change of frequency on a rather small value, the change in motional resistance is caused mainly by added weight. Therefore, one can assume that the β-casein layer was relatively rigid, which justifies application of the Sauerbrey equation.

After β-casein was bound to the surface, we could study its cleavage by different concentrations of chymotrypsin under flow condition for 35 min. An example of the changes of resonant frequency and motional resistance following the addition of 10 nM chymotrypsin are shown in Figure 10. In the presence of chymotrypsin, the resonant frequency increased by 35 Hz, but motional resistance decreased by 1.6 Ω. This is clear evidence of the cleavage of β-casein by chymotrypsin. Decrease of motional resistance can be due to an increase of molecular slip, which can be caused by weaker viscosity contribution.

Based on the changes of the frequency, we constructed a calibration curve for different concentrations of chymotrypsin (Figure 11a). For determination of LOD, we also prepared calibration curve at a low concentration range where the dependence was almost linear (Figure 11b). The LOD for gravimetric detection of chymotrypsin, 1.40 ± 0.30 nM was calculated from the Equation (2) using SD = 2.20 (%) and S = 5.17 (%) nM^−1^. This value was 2.8 times higher than that reported by ELISA methods and 9.3 times higher than for optical method of detection reported here. We should mention that gravimetric detection requires careful handling of the cell because even small changes in liquid pressure can affect the measurements. This method is, however, more tolerant to the presence of different components in the sample, such as added fat present in natural milk. The gravimetric method of protease detection requires time similar to optical methods. It is also important to mention the uniformity of the modified materials. The changes of frequency in gravimetric methods and absorbance maximum position in optical methods did not differ significantly from sample to sample, suggesting that it is not a source of significant error (large standard deviation).

It is also interesting to compare the K_M_ values determined in optical and gravimetric experiments. From the results presented above, it can be seen that K_M_ value in the case of AuNPs-based optical assay (3.89 nM and 1.03 nM for spectrophotometric and DLS methods respectively) were lower in comparison with those based on gravimetric measurements (K_M_ = 8.6 ± 3.6 nM, R^2^ = 0.999). This can be evidence of better access of β-casein substrate for chymotrypsin in AuNPs in comparison with those immobilized at the surface of piezoelectric transducer. This effect can probably explain the lower LOD for spectrophotometric method of chymotrypsin detection with those based on gravimetric method. In Table 1 we present comparison of other published methods for detection of chymotrypsin. Results obtained in our work have either comparable or higher LOD, but in most cases are faster in comparison to other methods.

**Table 1 biosensors-11-00063-t001:** The comparison of LOD and detection time of chymotrypsin detection.

Method	LOD	Detection Time	Reference
ELISA	0.5 nM	3.5 h	[10]
Liquid crystals protease assay	4 pM	3 h	[27]
Electrophoresis	20 pM	1 h	[28]
NIR fluorescent probe	0.5 nM	35 min	[29]
Ratiometric fluorescence probe	0.34 nM	30 min	[30]
UV-Vis, AuNPs	0.15 ± 0.01 nM	30 min	This work
DLS, AuMPs	0.67 ± 0.05 nM	30 min	This work
TSM	1.40 ± 0.30 nM	35 min	This work

## 4. Conclusions

We determined LOD for detection of chymotrypsin by gravimetric (LOD = 1.40 ± 0.30 nM) spectrophotometric (LOD = 0.15 ± 0.01 nM) and DLS method (LOD = 0.67 ± 0.05 nM). Spectrophotometric method showed the best value of LOD, even when compared to commercial ELISA (LOD = 0.5 nM). We also determined a steady-state constant K_M_ for the different methods with reverse Michaelis—Menten equation. The largest K_M_ value was found for gravimetric method of chymotrypsin detection (K_M_ = 8.6 ± 3.6 nM), followed by spectrophotometric method (K_M_ = 3.89 ± 1.24 nM),) and then DLS method (K_M_ = 1.03 ± 0.26 nM). We can explain observed differences in K_M_ values by difference in α-chymotrypsin activity, which is highest and least impeded on gold nanoparticles modified with β-casein. Addition of MCH decelerates the reaction and immobilization of β-casein on the gold surface slows the α-chymotrypsin ability to cleave β-casein. The detection time for methods that we tested was comparable and takes around 30 min for chymotrypsin determination. All methods required preparation of the sensing layers or modification of AuNPs overnight. The AuNPs or gravimetric sensors could be stored for a long time (more than one month) at 4 °C. In terms of difficulty in operation, the optical methods offered the easier way to measure chymotrypsin. With prepared AuNPs modified by β-casein and MCH, the spectrophotometric method required only one step of protease detection based on measurement of absorbance changes after 30 min, which was simpler in comparison with ELISA. DLS method based on AuNPs requires also only one step of measurement of the Z-average. The spectrophotometric method required only 50 μL of sample, the DLS method used 100 μL, while the gravimetric method used around 2 mL. One of the advantages of the gravimetric method is that it is more robust to “impurities” in the sample. The gravimetric method can be used with natural, no-transparent samples containing fat, minerals, or other proteins, just like in milk. Optical assays require a transparent sample; however, DLS method is a little less sensitive to changes in chymotrypsin concentrations. In terms of cost of analysis, the production of gold nanoparticles is relatively inexpensive and can be scaled to industrial amounts. For optical detection of chymotrypsin, gold nanoparticles should be surface-modified using inexpensive chemicals (β-casein and MCH). While gravimetric methods also use inexpensive chemicals for modification, but the cost of quartz crystal would raise the overall cost of the sensor. This cost offset can be reduced by multiple use of the same crystal when the sensing layer is regenerated. All methods have a distinct advantage and disadvantage compared to the currently used ELISA. In contrast with ELISA, the optical and gravimetric assay are not specific to the protease. Non-specificity of response can be addressed by using chymotrypsin-specific peptide substrate [13] or by integration of advanced machine learning algorithms [14]. In conclusion, we demonstrated advantages and disadvantages of spectrophotometric, DSL and gravimetric methods in detecting chymotrypsin. These methods can be applied also for detection of other proteases and can be useful for further application in the food industry and in medicine for real-time monitoring of the protease activity. In future work we plan to explore application of the presented techniques for analysis of natural milk samples (paying particular attention to gravimetric methods). Many new analytical methods use fluorometric or colorimetric molecules for detection of protease activity [31]. Gold nanoparticles seem to be good alternative component for colorimetric detection or for amplification of existing signal (for example increase of Raman signal from a sample using gold nanoparticles). It is clear that efforts furthering the development of new low-cost methods, easily implementable in practice, which would be sensitive, and exhibiting long-term stability, still need to be developed [32].

## Figures and Tables

**Figure 1 biosensors-11-00063-f001:**
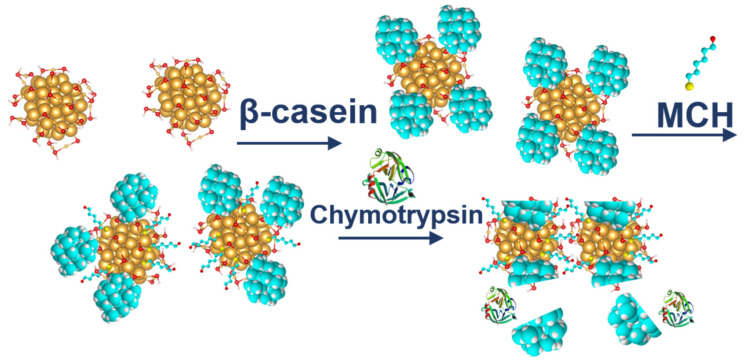
The scheme of gold nanoparticles (AuNPs) modification by 6-mercapto-1-hexanol (MCH) and β-casein and cleavage of β-casein by chymotrypsin.

**Figure 2 biosensors-11-00063-f002:**
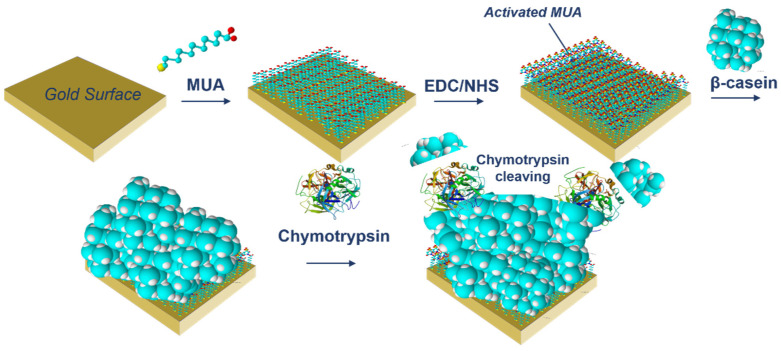
The scheme of modification of the piezocrystal and the cleavage of β-casein by chymotrypsin.

**Figure 3 biosensors-11-00063-f003:**
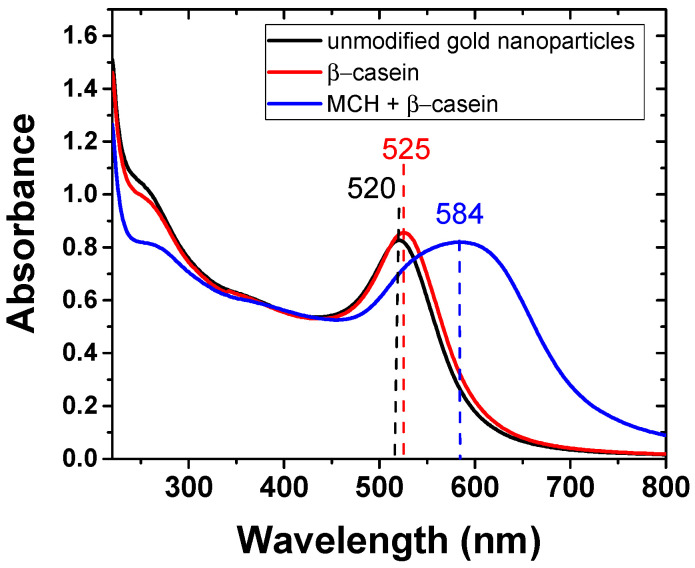
Absorption spectra of unmodified gold nanoparticles (AuNP) (black) and those modified by β-casein (red) and by β-casein + MCH (blue). The numbers at upper part of absorption peaks are wavelengths in nm.

**Figure 4 biosensors-11-00063-f004:**
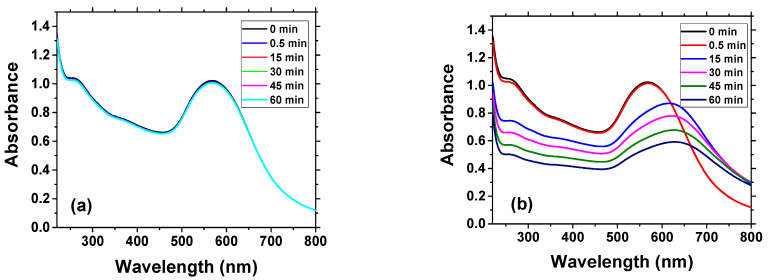
Changes of absorbance spectra of the suspension of AuNPs modified by β-casein and MCH in time for (**a**) 0.1 nM chymotrypsin and (**b**) 10 nM chymotrypsin.

**Figure 5 biosensors-11-00063-f005:**
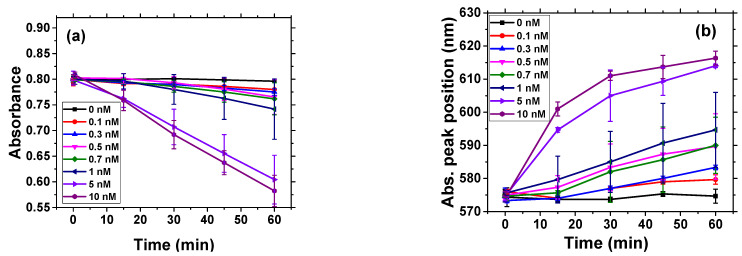
Changes in the absorbance (**a**) and in maximum position of absorbance peak (**b**) vs. time for different chymotrypsin concentrations in a suspension of AuNPs modified by β-casein and MCH. The results represent mean ± SD obtained from 3 independent measurements at each concentration of chymotrypsin.

**Figure 6 biosensors-11-00063-f006:**
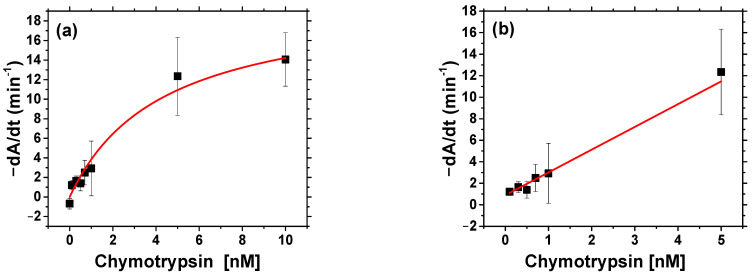
(**a**) Calibration curve for chymotrypsin fitted by reverse Michaelis—Menten model. dA/dt is numerical derivation of linear model of absorbance change at time *t* = 0 and corresponds to the rate of enzyme reaction. (**b**) Linear part of calibration curve −dA/dt vs. concentration of chymotrypsin for calculation of limit of detection (LOD). −dA/dt = (6.30 ± 0.23) × 10^−4^ min^−1^ nM^−1^ + (1.9 ± 0.4) × 10^−4^ min^−1^. (R^2^ = 0.993), LOD = 0.15 ± 0.01 nM.

**Figure 7 biosensors-11-00063-f007:**
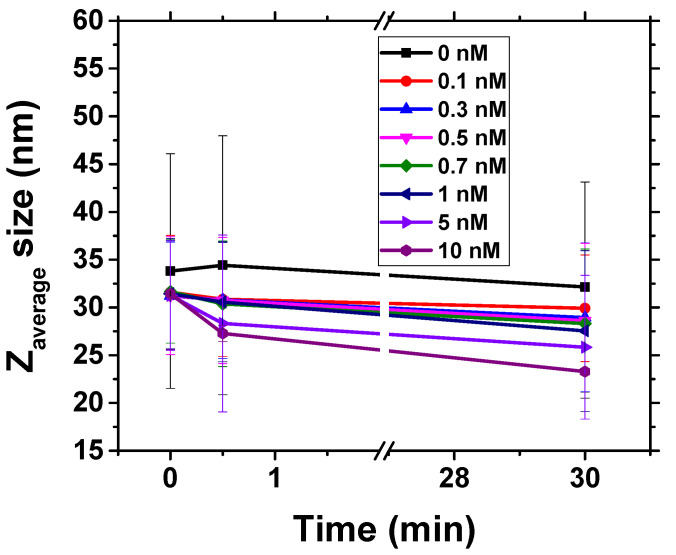
Change in Z-average size of AuNPs modified by β-casein at time 0 and 30 min at presence of various concentrations of chymotrypsin (see the insert). The results represent mean ±SD obtained from 3 independent measurements at each concentration of chymotrypsin.

**Figure 8 biosensors-11-00063-f008:**
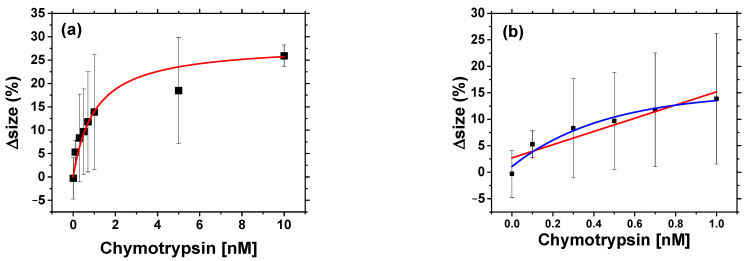
(**a**) The dependence of the changes of Z-average size (Δsize) vs. concentration of chymotrypsin for suspension of AuNPs modified by β-casein measured by DLS method. The curve represents fit according to reverse Michaelis—Menten model (see above). Δsize = [Z_average_(30 min)−Z_average_(0 min)]/Z_average_(0 min). (**b**) The linear part of calibration curve (red) used for calculation of LOD (Δsize = 12.47% nM^−1^ c + 2.71%, R^2^ = 0.87, where c is the concentration of chymotrypsin). For comparison we show first order reaction fit (blue color) (Δsize = 15.09%−13.83%e^−(c−0.006 nM)/0.45nM^, R^2^ = 0.998, where c is the concentration of chymotrypsin). The results represent mean ± SD obtained from 3 independent measurements at each concentration of chymotrypsin.

**Figure 9 biosensors-11-00063-f009:**
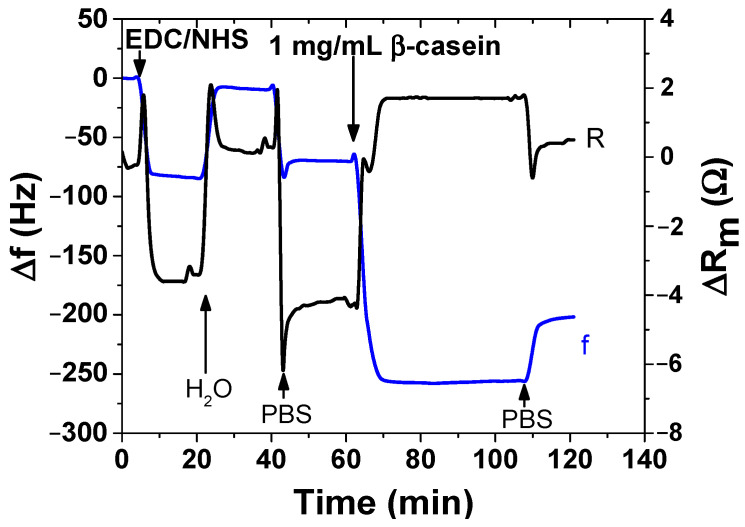
Kinetics of resonant frequency, f (blue), and motional resistance, R_m_ (black), changes during modification of piezocrystal by β-casein. The carboxylic groups of MUA that were chemisorbed at the crystal were first activated by EDC/NHS. The moments of addition of various compounds as well as washing the surface by water and PBS are shown by arrows.

**Figure 10 biosensors-11-00063-f010:**
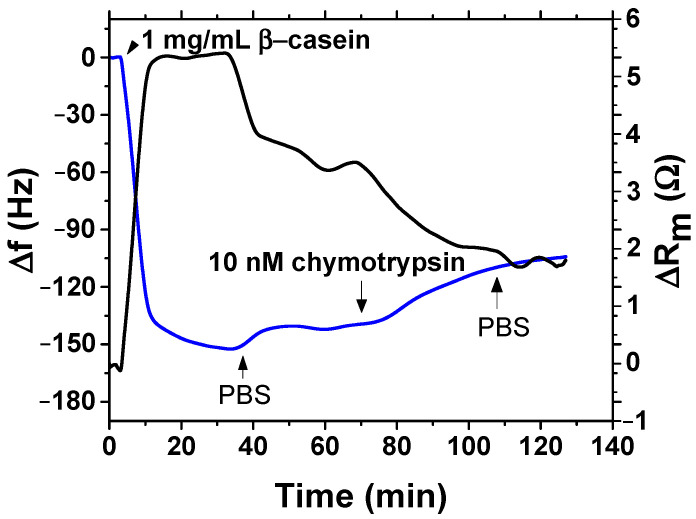
Kinetics of the changes of resonant frequency, f (blue), and motional resistance, R_m_ (black), following modification of piezocrystal by β-casein and addition of 10 nM of chymotrypsin. Addition of various compounds as well as washing of the surface by PBS is shown by arrows.

**Figure 11 biosensors-11-00063-f011:**
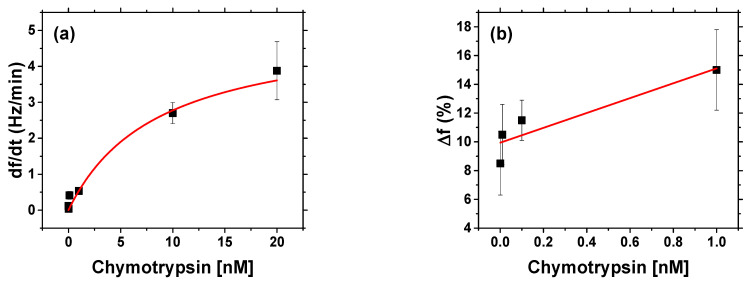
(**a**) Calibration curve for chymotrypsin fitted by reverse Michaelis—Menten model, v = df/dt was the first derivation of frequency obtained from kinetic curve. (**b**) Calibration curve: changes in frequency Δf = (Δf_chymo_)/(Δf_casein_) × 100 where Δf_chymo_ = f − f_0_ (where f is steady state frequency following addition of chymotrypsin and washing the surface by PBS and f_0_ those prior addition of chymotrypsin) is change in frequency after chymotrypsin cleavage and Δf_casein_ is frequency change after formation of β-casein layer. vs. lower chymotrypsin concentration range for determination of LOD. (Δf = 5.17%nM^−1^c + 9.94%, R^2^ = 0.84, where c is the concentration of chymotrypsin). The results represent mean ± SD obtained from 3 independent measurements at each concentration of chymotrypsin.

## Data Availability

Not applicable.

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
