# Peer review of "Detection of Chymotrypsin by Optical and Acoustic Methods"

_biosensors, 2021, doi:10.3390/bios11030063_

Round 1
Reviewer 1 Report
The authors provided a detailed comparison between spectrophotometric, DLS and QCM methods for chymotrypsin detection. The experiments were carefully carried out, the manuscript was well written and covered lots of practical aspects for sensor developments. Only minor edits are required before publishing:
- For broader readership, please add one or two sentences explaining the Michaelis-Menten reverse model, the information can be extracted, it's limitations, and include references.
- Please comment on the Km values of three methods.
- Please include the aspect of "cost" in the conclusion
Author Response
Please find authors responses in attachment

Reviewer 2 Report
Draft manuscript presented by Ivanov et al. is well organized, an overview of the methods for detection of chymotrypsin is included. Moreover, the Authors demonstrated advantages and disadvantages of proposed methods. The results are new and can be published. The manuscript can be published in Biosensors after minor changes.
- The authors should prepare Graphical Abstract to make the manuscript more attractive for potential readers.
- The abstract is written in a quite general way, the details about the analytical parameters and sensors major characteristics should be added.
- Introduction section should contain recent progress in the described field. Considering publications included in the Reference section it is not so obvious. More recent replacements should be found.
- The details on the calculation of the LODs should be added.
- Future perspectives should be presented more extensively.
Author Response

(The authors gave the same response as above.)

Reviewer 3 Report
The manuscript submitted to Biosensors entitled " Detection of Chymotrypsin by Optical and Acoustic Methods" by Ivan Piovarci et al. presents the preparation and detection studies of Chymotrypsin with three different methods.
In general, the authors did a good job on this research and the manuscript is written logically, with the overall claim well supported by the results. The structure of the article is simple, with a broad discussion about the subject supported by the description of several works in the literature about the methods, and the materials and methods are well described.
I don't have concerns about the scientific part and the results of the current study, are interesting. And only a few minor concerns. For instance:
a) Be careful with a few words that should be in italic throughout the manuscript text (e.g. et al. and b)
b) The introduction is too extensive. The information is important bur it should be resumed.
c) In figure 3, 4 and 10 the x axis is larger than the curve values. Please rectify.
Apart from this I don’t have any concerns, please check the manuscript carefully for typos and wording.
Thank you
Author Response

(The authors gave the same response as above.)

Reviewer 4 Report
Piovarci et al. measured the detectable limit of detection for Chymotrypsin using spectrophotometric, dynamics light scattering and quartz crystal microbalance. Compared to the previously reported ELISA, Liquid crystals protease assay, electrophoresis, NIR fluorescent probe, and ratiometric fluorescence probe, the detection time is similar or faster. The results performed by the authors are considered valuable as a new approach to the relevant field. Meanwhile, there are concerns about the following contents.
- figure 6, 7, 8 and figure 11B: In my personal opinion, the standard deviation is very large, and the author thinks the reason why the deviation is so large between samples. If there is no problem experimentally, isn't such a large deviation as a detection method that the error is severe? Meanwhile, the authors did not show any deviations from the absorbance values ​​in figures 3-5. Are these highly reproducible? Even so, I thinks that the standard deviation for Figure 5 should be added .
- As shown in Figures 1 and 2, the author has shown the experimental scheme well for the reader's understanding, and I think this is very useful for understanding manuscript. On the other hand, in terms of material, if the AuNPs modification by MCH and beta-casein, and piezocrystals are not made uniform, the experimental results are expected to be different. How can the author be sure that the fabricated materials are made uniformly?
- Authors should clearly state the catalog number for the product used in the experiment, because there are various type of beta-casein and alpha-chymostrypsin from sigma-aldrich, and they have various activities and purity. In addition, the authors mentioned that alpha-chymotrypsin used in the experiment was the highest purity. Specifically, what is the purity, and it is questionable whether they can be used for experiments without further purification. I think that it is judged that their activity or aggregation in the sample may affect the activity.
- Information about the buffer for alpha-chymotrypsin is missing from the experimental method. What buffer was used to dissolve alpha-chymotrypsin and beta-casein, and what buffer was used for enzyme reaction. And the buffer used is the optimal condition for alpha-chymotrypsin activity?
minor
- line 156-157: "... 0.01% chloroauric acid (HAuCl4) has been boiled and then 5 mL of 1% sodium tris-citrate was added. This solution was continuously boiling ..." At what degree did it boil, and is there no problem with the change in concentration due to the moisture evaporated during this process?
- Is the concentration of alpha-chymostrypsin calculated considering purity? How was the purifity checked and how was the concentration measured?
Author Response

(The authors gave the same response as above.)

Round 2
Reviewer 4 Report
Thank you for the author's sincere response. I suggest adding responses to comments 1, 2, and 4 to the discussion section. This may be negative from the manuscript research point of view, but I think it is an improvement point in future related studies.
Author Response
Reviewer 4
We are grateful to the reviewer for additional most useful comment, which has been addressed in the revised manuscript. The changes or additions are highligted by yellow..
Comment: I suggest adding responses to comments 1, 2, and 4 to the discussion section. This may be negative from the manuscript research point of view, but I think it is an improvement point in future related studies.
Response: We added responses to comment 1 to page 9 lines 298-303, to comment 2 to page 11 lines 377-381 and comment 4 to page 9 lines 303-307